# A science impact framework to measure impact beyond journal metrics

**Mary D. Ari** [ID]*, **John Iskander, John Araujo**¤**, Christine Casey, John Kools, Bin Chen, Robert Swain, Miriam Kelly, Tanja Popovic**

Centers for Disease Control and Prevention (CDC), Atlanta, Georgia, United States of America

¤ Current address: Drug Enforcement Administration, Department of Justice, Arlington, Virginia, United States of America

* Mkb9@cdc.gov

**Data Availability Statement:** All relevant data are within the manuscript in Tables 1 and 3.

**Funding:** The author(s) received no specific funding for this work.

## Abstract

Measuring the impact of public health science or research is important especially when it comes to health outcomes. Achieving the desired health outcomes take time and may be influenced by several contributors, making attribution of credit to any one entity or effort problematic. Here we offer a science impact framework (SIF) for tracing and linking public health science to events and/or actions with recognized impact beyond journal metrics. The SIF was modeled on the Institute of Medicine's (IOM) Degrees of Impact Thermometer, but differs in that SIF is not incremental, not chronological, and has expanded scope. The SIF recognizes five domains of influence: disseminating science, creating awareness, catalyzing action, effecting change and shaping the future (scope differs from IOM). For public health, the goal is to achieve one or more specific health outcomes. What is unique about this framework is that the focus is not just on the projected impact or outcome but rather the effects that are occurring in real time with the recognition that the measurement field is complex, and it takes time for the ultimate outcome to occur. The SIF is flexible and can be tailored to measure the impact of any scientific effort: from complex initiatives to individual publications. The SIF may be used to measure impact prospectively of an ongoing or new body of work (e.g., research, guidelines and recommendations, or technology) and retrospectively of completed and disseminated work, through linking of events using indicators that are known and have been used for measuring impact. Additionally, linking events offers an approach to both tell our story and also acknowledge other players in the chain of events. The value added by science can easily be relayed to the scientific community, policy makers and the public.

## Introduction

Many frameworks have been used to measure programs, research, and other aspects of science and technology advancements [1–7]. Commonly used measures of science and research impact often are based on publication metrics [3]. There has been heavy dependence on quantitative measures by the scientific community, driving the value of journal metrics, with

**Competing interests:** The authors have declared that no competing interests exist.

various indices having been developed to credit publication contributions to knowledge [3, 6]. This is not unusual as scientific, peer-reviewed publications are recognized as one of the most important formal outputs or deliverables of a research project that can be used to infer the quality and impact of the underpinning science. In addition, journal metrics, such as citations and impact factor are relatively easy to collect, and they are valuable indicators of the reach of the research in terms of how widely it is disseminated and its uptake. But they do not characterize the influence created such as resulting actions or changes or the way in which the research knowledge is used.

Funders continue to grapple with how to assign measurable criteria of a more practical value to research under review in research proposals and awards [8–10] or said another way, the impact of science and research efforts beyond just the publication of findings [4, 11], because using a metrics-only approach will not suffice to capture broader societal impacts on economic, technologic and innovative advancements [2]. While this idea is welcomed by some, others express reservation driven by the concern that innovative research may be stifled this way [12, 13]. How and when to use these measures is a subject of intense debate [3, 10, 14].

While CDC has a framework for program evaluation in public health that is widely being used for public health programs [15], this evaluation framework has not been conducive for assessing the impact of science and research efforts. Other frameworks used for evaluation of public health interventions are mostly very specific and narrow in scope, limiting broad applicability [16–19]. For example, a framework such as Reach, Effectiveness, Adoption, Implementation and Maintenance (RE-AIM), is narrowly focused on evaluating behavior change in health interventions, it does so effectively, and has been adapted for use in evaluating built environment strategies [20]. However, it is only flexible enough to be applied to the evaluation of similar applications within the scope of its design. Instead, we are turning our attention to a broader assessment of how to describe the role of science in contributing to the improvement of public health, for which we developed the Science Impact Framework.

## Materials and methods

### Developing the science impact framework (SIF)

To develop this framework, a literature review was undertaken to identify frameworks previously developed or used [1–7, 21]. Next, we considered those elements of the frameworks identified that would best demonstrate the impact of CDC science. One example of this is the Payback Framework which has been in existence since the 90's and is applied to medical and health services. Several other frameworks developed later are based on the Payback Framework [2, 3]. The Research Excellence Framework (REF) and the Research Quality Framework (RQF) emerged recently. The existing frameworks we studied are mostly research or health services frameworks. In order to capture other science efforts, such as developing guidelines and recommendations that contribute to health outcome, we defined science more broadly than research. We embraced some of the concepts we highlighted from these frameworks (Table 1). But as our primary model, we adapted the Institute of Medicine (IOM) "Degrees of Impact" Thermometer [21]. The key attraction of the IOM model was the focus on influences. However, we needed to extend and expand these concepts because IOM serves in an advisory role so, the scope of their work is in the realm of knowledge diffusion (user- pull end of spectrum), while CDC has a broader scope; diffusion of knowledge, applied research, technology creation, capacity building, and program/initiative implementation. And the IOM model suggests an incremental progression of processes and actions, our model fundamentally differs in this aspect as well.

Table 1. Description of impact frameworks reviewed.

| Reviewed Frameworks (Description) | Description | Elements incorporated into the Science Impact Framework |
|---|---|---|
| Payback Framework | The Payback framework has been in existence since the 90's and is applied to medical and health services. It focuses on knowledge, production, capacity building, informing policy, broader societal and economic impact. | It addresses diffusion of knowledge and goes further, and the SIF embraced looking deeper into how the Knowledge is used. The focus areas were also considered in generating the SIF key indicators |
| United Nations Development Program | A development framework that tracks societal advancement using economic, technology, innovation and human skills as indicators. It has what is called the technology achievement index that is used to rate nations. | Elements of societal advancement were added to the SIF key indicators, SIF did not include the idea of rating as we think that will detract from our key focus which is showing impact. |
| Research Excellence Framework (REF) | The REF is used by the United Kingdom higher education funding bodies it has three elements which are weighted; output 65%, impact 20%, environment 15%. The impact element assesses the quality of the research and the ability to demonstrate benefits to the wider economy and society. Case studies are used to demonstrate impact. This is retrospective and captures impact between 2008–2013 for research with some output 15 years prior. | Using case studies to demonstrate impact is useful especially for retrospective studies that involve large body of work and covers several years. |
| Research Quality Framework (RQF) | The RQF is the Australian health services impact framework. It was developed to measure impact of primary health care research. The focus is on outcomes or measures of uses of the research knowledge in other research, in policy or service. In addition, drivers of impact, that is, producer push (dissemination) or user pull (uptake), are tracked. | SIF incorporated the tracking of producer push as dissemination and user pull as creating awareness. |
| Institute of Medicine (IOM) "Degrees of Impact" Thermometer | The IOM framework elements are: Spreading the Message; Receiving Recognition; Informing the Field; Inspiring Action; Effecting Change. | The key attraction of the IOM model is the focus on influences. |

## Description of the science impact framework

The Science Impact Framework (SIF) consists of five domains of impact each with key indicators for the specific domain (Table 2).

The resulting SIF is a collection of logically related or associated elements (influence). Influence in this case is the term used to describe the evidence of impact within each domain of the SIF as described by the key indicators. Description of the domains of impact are as follows:

1. **DISSEMINATING SCIENCE**: This represents producer push and may include the publication of findings in peer reviewed journals or other reports, presentations at conferences or through other media channels.

2. **CREATING AWARENESS**: This represents user pull and may include awards, general awareness, or acceptance of a concept or findings by scientific community or policy makers, generating new discussion based on shared science.

3. **CATALYZING ACTION**: This represents actions taken as a result of the science and may include partnerships and collaborations, technology creation, new funding, congressional hearings or bills, or introduction in practice.

4. **EFFECTING CHANGE**: This represents changes that occur as a result of the science or the actions taken, and may include building public health capacity, legal/policy change, cultural/social/behavior change, or economic change.

5. **SHAPING THE FUTURE**: This represents additional considerations (scope beyond IOM "Degrees of Impact" Thermometer) that affect the future direction, drive further progress in understanding of the science, or implementation in practice, and may include new hypothesis or strategies, implementation of new programs/initiatives, or quality improvement.

**Table 2. The science impact framework–Examples of indicators and data sources*.**

| Domain of Impact (Description) | Examples of Data Sources | Potential Measurable Indicators |
|---|---|---|
| **Disseminating Science** (Generating and communicating knowledge by the producer) | Investigators, MEDLINE®, Web of Science, Google, Google Scholar | Scientific publications (open access journals), trade publications, professional meetings/conferences, general communication (social media, Web, print), presentations, training, coursework, other scientific output |
| **Creating Awareness** (Uptake of knowledge and further dissemination and dialogue by the user) | Investigators, LexisNexis®, Web | Continuing Education (CME, CEU), recognition awards, stakeholder resources, curriculum, and training, feedback (survey, focus groups, anecdote), information sharing and communications among professional societies, electronic communications (information shared on listservs and other electronic resources, social media, news coverage), queries, requests to contribute to efforts that further the science output |
| **Catalyzing Action** (Adoption of knowledge resulting in specific actions) | Investigators, registries (patents, trademarks), marketing, legislation | Technology creation, new funding (pilot studies/research), advocacy groups/non-governmental organizations, congressional hearings, partnerships and collaborations, research and development, office practice/point of care changes |
| **Effecting Change** (Changing current or existing situations, directions, strategies, policies, or practice) | Investigators, surveillance systems, guidelines and recommendations (G&R) | Building public health capacity (e.g., workforce development, funded research, improved staff competency), creation of registries/surveillance, legal/policy changes, accreditation, cultural/social change, behavioral change, economic change, change instilled, new/formal guidelines and recommendations (e.g., World Health Organization (WHO), hospital standards, funding, anecdotes/case studies, sustainable and scalable science translation |
| **Shaping the Future** (Implementing new or furthering improvements and changes) | Investigators, surveillance systems, G&R | New hypotheses/continuous quality improvement, implementation of new public health programs/initiatives |

*Data sources include a mixture of stakeholders (who are experts for identifying data sources), systems (that can provide the data), and actual measures (e.g., patents, trademarks, or guidelines and recommendations). This is an abbreviated list of data sources; other resources as they become available may be used as needed.

Fig 1: The Science Impact Framework, illustrates the SIF with five domains of scientific impact that express the scope and type of influence generated by the scientific undertaking. The degree of impact is not necessarily a linear sequence of progression through the five domains; therefore, events captured may not be reflected in every domain and may not occur chronologically. The model also portrays the complexity of the measurement environment with other influences beyond the ones described by the domains of the SIF. For example, there may be other influences that may or may not work synergistically with the desired influence for the work under consideration. Thus, impacting the ability to achieve desired outcome positively or negatively. Our model uses both quantitative and qualitative

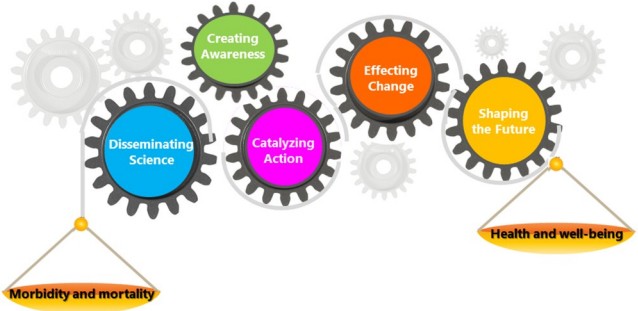

**Fig 1. The science impact framework.** Health outcomes are the ultimate goals, driven by the 5 domains of influence as shown in Table 1. Health outcomes, for example include positive effects on prevalence and incidence (e.g., frequency of outbreaks, trends); reduction in morbidity and mortality; increased life expectancy; and increased quality of life improvements.

measures. The types of impact of interest transcend from the impact on the field, to broader societal impacts, including policy and practice impact and for CDC, the goal is health outcomes.

For the purpose of applying the SIF, we define CDC science broadly to include 1. basic and applied epidemiology, 2. laboratory studies, 3. Surveillance, and 4. other scientific outputs such as models, methods, meta-analyses and guidelines and recommendations developed to improve prevention and control or improve the practice of public health. The tracing and linking of actual instances of scientific influence through the framework involves either identifying points of impact and tracing backward events related to the science or research impact or going back to original scientific work (which may include synergistic efforts) and tracing forward events that have link to that work [5]. The latter approach ensures clear linkages can be made, and it is feasible to identify effects within 2–5 years of dissemination since these effects may be formative and do not have to be the ultimate outcome. The SIF relies on user judgment as expert opinion, which could be supported by peer reviews, or interviews, to identify credible links that can be traced through the framework. This is how it works:

The reviewer identifies a point of scientific significance and places it within one of the SIF five domains of influence based on alignment with key indicators for that domain. The reviewer using the framework further:

- Identifies forward and/or backward events or activities that link or can be associated (logically or empirically) with the point of scientific significance,

- Validates the links with peer review (which could be internal or involve external partners) or expert opinion,

- Assigns or reassigns the linked events to one of the five domains of influence.

Because the SIF can be used to track the impact retrospectively or monitor it prospectively, it is a culture change from focusing on outputs or journal metrics. It allows the investigating of what changes occurred or are occurring because of the work. Ultimately, the impact can be tracked to an individual, groups or entire society.

## Validating the framework using case studies

Once the SIF was developed it was evaluated for functional utility using publications of research and findings that cover various areas of public health: basic research, laboratory science, epidemiology, guidelines and recommendations, surveillance, infectious diseases, non-communicable diseases, meta-analyses, and evaluation (Table 3). These topical areas were selected from a) *Morbidity and Mortality Weekly Report (MMWR)* published by the CDC, b) papers competing for CDC's Charles C. Shepard Science Award (https://www.cdc.gov/od/science/aboutus/shepard/) [22], and c) topics from CDC's Public Health Grand Rounds (PHGR), monthly webcasts that addresses key public-health challenges [23]. The data for tracing of influence in each domain using identified key indicators included citation analysis and subject matter experts' opinions (Table 2), a combination of these sources was used to establish effects and linkages. In addition, current journal metrics in use for measuring impact such as number of citations, impact factor, were also assessed for each of the original manuscripts of our case studies to compare with influences and impact identified by the SIF model.

In addition, a case study approach was used to test the SIF, a total of 11 case studies were conducted (Table 3). The starting point for each case study was a publication describing the findings or output of interest. Three of the eleven case studies consisted of more than one relevant publication as starting point. Specifically, the tuberculosis (TB) and Group B streptococcal

**Table 3. Case studies used to validate the science impact framework.**

| Case Studies* | Brief Description of Publication (S) | Years** | Areas of Public Health | Summary of Key Impact Findings Using SIF |
|---|---|---|---|---|
| **Blood Alcohol Concentration** [24] | Results of systematic reviews are presented. There was strong evidence for the effectiveness of the 0.08% blood alcohol concentration (BAC) laws, minimum drinking age laws, and sobriety checkpoints in reducing injury and deaths due to alcohol-impaired driving. | 10 | Non-communicable disease- meta-analysis | Disseminated findings resulted in 0.08% BAC laws approval by Congress and signing into law by President Clinton in 2000 [24, 25] (Catalyzing action). Within 4 years all 50 states had implemented the law, and self-reported drinking and driving episodes declined by 49 million in 2006–2010. By 2009 alcohol impaired driving deaths declined to <11,000 from >13,000 recorded in 2006 National Highway Traffic Safety Administration [26, 27] (Effecting change). |
| **Tuberculosis (TB) Screening and Prevention for People Living with HIV (PLHIV)** [28–30] | Improving diagnosis of TB in people with HIV" study demonstrated that a simple, sensitive approach using 3 symptoms could rule out TB "CDC-Botswana isoniazid preventive therapy (IPT) Trial 2004–2009" demonstrated 36 months of IPT was superior to the internationally recommended 6 months of IPT to prevent TB in people with HIV for whom TB disease had been excluded | <1–2 | Infectious disease-laboratory science | Disseminated findings resulted in a policy and practice change in all 3 countries involved in the studies [31]. In addition, there was further meta-analysis by World Health Organization (WHO) (Catalyzing action), and subsequent changes to its recommendations for TB screening and IPT for PLHIV in TB endemic settings [32] (Effecting change). |
| **Guidelines for Field Triage of Injured Patients** [33] | In 2005 CDC facilitated the revision of the field triage guideline. The rationale for the 2006 revision of the field triage criteria is described. The Field Triage Decision Scheme (Decision Scheme) serves as the guide for emergency medical services (EMS) providers to determine the most appropriate destination facility with the use of four decision steps (physiologic, anatomic, mechanism of injury, and special considerations). | 2 | Non-communicable disease- Guidelines and recommendations | A 2-year prospective observational study of 11,892 patients at 3 Level 1 trauma centers indicated that use of the 2006 Guidelines would have resulted in identifying 1,423 fewer patients for transport to a trauma center at the expense of 78 patients being under-triaged [34]. Another study estimated national savings of $568 million per year [35] (Catalyzing action). The Guidelines have been adopted by the National Registry of Emergency Medical Technicians and endorsed by the Federal Interagency Committee on Emergency Medical Services (FICEMS) [36] (Effecting change). |
| **Healthcare-Associated Infections (HAI)** [37] | Healthcare Infection Control Practices Advisory Committee (HICPAC) developed recommendations to guide legislators on mandatory healthcare-associated infection (HAI) reporting. HICPAC recommended that persons who design and implement systems (a) use established public health surveillance methods, (b) create multidisciplinary advisory panels, (c) choose appropriate process and outcome measures, and (d) provide regular and confidential feedback of performance data to healthcare providers. | 6 | Infectious disease-Guidelines and recommendations | Congress addressed HAI prevention as part of the Affordable Care Act (ACA) by requiring hospitals to report Central Line Associated Blood Stream Infections (CLABSIs) in ICU patients beginning January 2011 via the National Healthcare Surveillance Network [38] (Shaping the future). As of 2011, 28 states had passed laws mandating public HAI reporting. Twenty-two of the 28 states specified that HAI reporting occurs through CDC's National Healthcare Safety Network (NHSN), increasing the number of hospitals from 300 in 2006 to >3,500 in 2010 [38, 39] (Effecting change). |
| **Pediatric Cough and Cold Medications (CCM) as a Sentinel for Pediatric Medication Overdoses** [40, 41] | The National Association of Medical Examiners and the CDC investigated cough and cold medication (CCM) deaths of U.S. infants' ≤12 months of age. The study reported 3 infants' ≤6 months of age who died from cough and cold medications. The blood levels of pseudoephedrine found in the 3 infants were approximately 9–14 times the levels resulting from recommended doses for children 2–12 years of age. The study showed nationally representative morbidity data about age-specific adverse events from cough and cold medications, associated emergency department visits, with unsupervised ingestions to be the frequent cause of adverse events. | 3–4 | Infectious disease -surveillance | In October 2007, an FDA advisory committee recommended ending use of CCMs for children <2 years old. Subsequently, the Consumer Healthcare Products Association (CHPA) announced a nationwide, voluntary withdrawal of >300 CCMs labeled or marketed for children <2 years old [42] (Catalyzing action). A 50% reduction in emergency department (ED) visits for CCM ingestions among children <2 years was reported after the withdrawal of CCM marketed for infants [43]. In 2010, a 10% reduction in ED visits for medication overdoses among children less than 5 years of age set as a U.S. Healthy People 2020 objectives [44] (Effecting change). |

*(Continued)*

**Table 3.** (*Continued*)

| Case Studies* | Brief Description of Publication (S) | Years** | Areas of Public Health | Summary of Key Impact Findings Using SIF |
|---|---|---|---|---|
| **Changes in Medicaid Physician Fees and Patterns of Ambulatory Care** [45] | Cuts in Medicaid physician fees led to statistically significant reductions in the number of visits for Medicaid patients compared to privately insured patients. In addition, there was a shift away from physician offices and toward hospital emergency departments and outpatient departments, particularly for (a) hypertension, (b) asthma, (c) urinary tract infections, and (d) diabetes. | 2 | Evaluation | Changes in Medicaid physician fees and patterns of ambulatory care case study showed that a similar study was conducted in the Republic of Georgia to examine the responsiveness of private providers to beneficiaries of the Medical Insurance for the Poor (MIP) [46] (Creating awareness), the paper was cited in decision to deny the state of California's proposal to further decrease provider payment rates [47]. (Catalyzing action). |
| **Risk of Bacterial Meningitis in Children with Cochlear Implants** [48] | The incidence of meningitis caused by *Streptococcus pneumoniae* in cochlear implant recipients (children < 6 years of age when they received the implant) was more than 30 times the incidence in a cohort of the same age in the U.S. patients who received an implant with a positioner had a higher incidence of meningitis than those who did not have the positioner. | 8 | Infectious disease-epidemiology | A new animal model that will allow risk assessment of meningitis post cochlear implant was developed [49]. CDC and the Advisory Committee on Immunization Practices (ACIP) recommended that all individuals with cochlear implants receive age-appropriate vaccination against pneumococcal disease as recommended for other persons at high risk for invasive pneumococcal disease CDC [50] (Catalyzing action). During the Pneumococcal Conjugate Vaccine 7 (PCV7) shortage in 2004, children with cochlear implants were identified among the high-risk children that should still receive the 4th dose of PCV7 CDC [51] (Effecting change). |
| **West Nile Virus Vaccine** [52] | A single intramuscular (IM) injection of a pCBWN DNA vaccine could prevent West Nile Virus (WNV) infection in mice and horses. The plasmid vaccine could be used to produce viral antigens useful in WNV diagnostics. | 10 | Infectious disease-basic research | In 2002, the California Condor Recovery Team learned that an experimental DNA WNV vaccine protected against WNV infection in several bird species. CDC expedited the delivery of the equine WNV vaccine. In 2004, the vaccine was found to be safe and effective in protecting captive condors from naturally circulating WNV [53]. (Catalyzing action). In 2005, the CDC equine DNA vaccine was licensed by the USDA. This led to a Phase 1 human clinical trial of a similar DNA vaccine shown to induce T-cell and antibody responses at levels shown to be protective in studies of horses [54]. (Effecting change). |
| **Decline in Invasive Pneumococcal Disease** [55] | The rate of invasive pneumococcal disease (IPD) dropped from an average of 24.3 cases per 100,000 persons in 1998 and 1999 to 17.3 per 100,000 persons in 2001 with largest decline in children. The use of the pneumococcal conjugate vaccine (PCV7) has prevented disease in young children and may possibly reduce the rate of disease in adults. | 8 | Infectious disease-surveillance | The PCV7 vaccine prevented more than twice as many cases of IPD through indirect effects [56] (Catalyzing action). By 2007, IPD among U.S. adults fell by over 90%, and most of these individuals had not been vaccinated, showing the benefit of herd immunity [57, 58] (Effecting change). |
| **Prevention of Perinatal Group B Streptococcal (GBS) Disease** [59–61] | In 1996, CDC, in collaboration with relevant professional societies, published guidelines for the prevention of perinatal group B streptococcal disease. Those guidelines were updated and republished in 2002. In 2010 CDC updated the guidelines which included the following key changes that included expanded recommendations on laboratory methods for identification of GBS. | 1 | Infectious disease-Guidelines and recommendations | The guidelines for intrapartum chemoprophylaxis to reduce GBS have helped reduce rates of early onset infection but reflect a continued burden of disease [62] (Catalyzing action). Nationally representative hospital discharge diagnostic code data demonstrated a steady decrease in clinical sepsis rates during 1990–2002, with a marked decline in clinical sepsis among term infants following the issuance of the 1996 GBS prevention guidelines; these data suggest that the observed decline in early-onset GBS disease is a result of prevented cases of illness and not simply of sterilization of neonatal blood cultures as a result of exposure to maternal antibiotics [63] (Effecting change). |

(*Continued*)

**Table 3.** (Continued)

| Case Studies* | Brief Description of Publication (S) | Years** | Areas of Public Health | Summary of Key Impact Findings Using SIF |
|---|---|---|---|---|
| **Use of WHO and CDC Growth Charts for Children** [64] | The rationale for the use of the growth charts from the World Health Organization (WHO) and CDC was described. Specifically, it was recommended that the 2006 WHO international growth charts be used for children <24 months of age and the 2000 CDC growth charts for persons 2–19 years of age. It was noted that the CDC growth charts served as standards rather than references. This recommendation recognized that breastfeeding is the recommended standard for infant feeding and that screening for abnormal or unhealthy growth should use the 2.3rd and 97.7th percentiles. | 1 | Maternal and child health- Guidelines and recommendations | In 2006, CDC, the National Institutes of Health, and the American Academy of Pediatrics (AAP) convened an expert panel to review scientific evidence and discuss the potential use of the new WHO growth charts in clinical settings in the United States. Based on input from this expert panel, CDC recommended that clinicians in the United States use the 2006 WHO international growth charts, for children aged <24 months and the 2000 CDC growth charts for persons 2–19 years of age [64]. CDC made software for the WHO charts available in SAS and Visual Basic (Catalyzing action). USDA's WIC program, which serves over 50% of new births in the U.S. has enacted policy that the WHO charts should be used for the assessment of growth in children under 2 [65] (Effecting change). |

*Case studies are retrospective, and the starting point for each study was original manuscript (s) tracing forward to 2011.

**Years since publication through 2011

(GBS) case studies had three, and the pediatric cough and cold medication (CCM) had two relevant publications respectively, (Table 3). Findings from these publications worked in synergy with each other to produce the documented impact. Citations of the original publications were identified, and each reviewed to assess and understand the way the disseminated knowledge or output was used in the citing publication. In the 11 case studies that were used, the time frame assessed was from time of dissemination of scientific knowledge and tracing it forward to 2011 (Table 3). This is like systematic review; in this case, qualitative analysis was undertaken to investigate evidence of importance of the original published work. Identified effects or influences were placed under the relevant domain of influence as described by the SIF. When tracing the events, it was important to research in more detail the role/influence of the original CDC manuscript(s) in these events to establish documented links between the manuscript(s) and these events and identify a link to the that domain of influence based on alignment with key indicators for the domain. Not all domains of influence are utilized, and the link does not have to be to the immediate domain as listed in the SIF model. Appropriateness of links and placement of events under each domain is validated through expert opinions.

## Results and discussion

### Summary of key findings from case studies

A summary of key findings using SIF are presented in Table 3. Bibliometric analysis was done on the original manuscript(s) to show the number of primary and secondary citations. Naturally, the number of citing sources was minimal when an original paper was recently published and when the topic may only be of professional interest to a narrow audience. A few of the case studies have been presented at the CDC PHGR [66] https://www.cdc.gov/grand-rounds/pp/2014/20140121-science-impact.html and at the Office of Science web page https://www.cdc.gov/od/science/impact/testing.html.

**Further considerations based on the case studies.** It is important to disseminate findings through publications, but that does not represent the end-product of research, rather the beginning of further influence. Impact beyond publications could be in form of products and technology; however, we recognize that ultimately information about these products,

programs, initiatives and advancements can be provided in the form of publications, thus dissemination of science was used as one of the domains of influence in the SIF. Publication metrics such as number of citations would likely underestimate the impact of the work. A careful review of citations data for our case studies suggest the community of users drives the number of citations. For example, a comparison of publications from two of our case studies (Table 3): The publication on cochlear implant had 97 citations, 761 2nd generation citation, 5-year impact factor of 52.36, and 9.70 average cites per year, versus the publication on pneumococcal vaccine, which had 1,035 citations, 23,415 2nd generation citations, 5-year impact factor 52.36, and 103.6 average citations per year. These were published in the same journal, the same year, and consequently have the same impact factor; yet, there is a significant difference in both the primary citation and second-generation citations. Hence, impact factor of the journal does not seem to be the driving factor. The size of the community that needs the science or information can vary significantly and therefore can influence the number of citations. For example, the cochlear implant publication will be of interest predominantly to manufacturers of the device, physicians, and patients who use them, and as a result, the numbers are small compared to infectious disease such as pneumococcal pneumonia that affect significantly larger number of people. Merely counting citations does not reveal the way the science was used e.g. as background information or foundational to the steps or actions taken. Furthermore, just because an article is cited does not mean it is for a positive reason, sometimes articles are cited as examples of bad or flawed science [67, 68]. There is ample evidence that even publications that have been retracted as bad science or due to scientific misconduct continue to get cited [69]. Newer measures, such as Altmetric have similarly been found not to reflect broad societal impacts [70] but can provide data on the reach of publications and be a good resource in using the SIF. The main reason all the afore mentioned indicators are attractive is that they are quantitative and readily available. Perceptions of participants in a recent evaluation suggest that the incentivizing of publications may be at the expense of generation of broader impacts [71]. Scientific work is generally not linked to dollar investment or time to produce results. However, assessment using the SIF would prompt the question—Was the investment of dollars, time and efforts worth it? Just because a publication is infrequently cited does not discount the potential magnitude of contribution. For example, in the West Nile case studies (Table 3), the bibliometrics of this publication [52] showed 193 citations. However, the findings were instrumental to development of animal vaccine and subsequently human vaccine [66] (https://www.cdc.gov/grand-rounds/pp/2014/20140121-science-impact.html).

The body of work with impact on practice and policy especially, is rarely captured in peer reviewed publications. Currently, there is no easy way to get to these types of information. Most of the citations are peer reviewed journals with a few books and conference proceedings. Hence, impact on practice and policy is rarely captured. The CDC scope of work as the United States premier public health agency, leads to technology creation such as laboratory methods, analytical methods and in addition, the knowledge generated from research informs further actions such as policy, practice and future research. In our search for information related to key indicators that define each domain of influence in the SIF, we found that the peer review literature was not necessarily the best venue for the information we sought. It took a combination of discussions with subject matter expects and internet searches for progress in the topical area to identify non-quantitative measures. Indicators that are qualitative in nature are more difficult to find without a deliberate effort and having a system in place that captures such information. Examples include policy changes, ongoing dialogue, and changes in practice.

It is important to measure the broader impact of CDC science on research, technology, practice and ultimately health outcome. Interest in evaluating science and research arises as both an interesting problem in scholarship and for public value. The challenge of scholarship

lies in the complex environment of the science and research enterprise as well as how knowledge is accumulated and disseminated. Making these relationship even more complicated is an agency's portfolio of science and research in which more than one project yields results that are commingled into a single output without clear specification or linkage of how to attribute the contribution of the various lines of science and research into the combined output [72].

## Applicability of the science impact framework

The issue of how to measure impact of science or research is not simple and is even more complex for public health science. Measurement, especially when it comes to health outcomes, is complex because, it could take years, multiple actions may be involved, multiple players may be involved, which raises the issue of where to assign the credit. Because impact is frequently the result of synergy of many factors, it is important to not overestimate a single contributor; highlighting importance of collaborators can lead to stronger partnerships; professional networks have been shown to be effective in promoting uptake of research findings [73]. Because there are other players, other strategies and approaches as well, that may or may not be synergistic, the SIF allows a view of positive or negative effects.

Finally, decision on what to measure that will provide the most value, and the venue to obtain data for such measurement is a challenge. Systems with the capacity to capture all interactions including outputs and interim impact have been suggested as a possible approach since those provide a network of data [74]. The clear delineation of potential domains of science influence inherent in the SIF provides a useful construct. It helps science initiatives to be viewed through the lens of practice, and as a result, ask and answer similar kinds of questions as more traditional implementation efforts regarding what impact can be made or is being made and what changes will produce bigger or yield most distal outcomes. Therefore, the SIF complements and strengthens traditional evaluation [15]. The SIF serves both planning and evaluation functions; it helps us think about the myriad of outcomes that result from science and research efforts and that singly or jointly allow our efforts to contribute in the longer-term to improve health outcomes. The SIF provides a very useful description of the sequence of potential outcomes for a science effort without assuming that the same sequence will hold for all science efforts or that all science efforts will affect the most distal health outcomes in the framework. As such, it allows for constructive discussions with stakeholders and skeptics alike who wonder about the extent of our accomplishments where the direct relationship to health outcomes is difficult to demonstrate. Likewise, by laying out an expected sequence of outcomes for an effort, it is possible to look for those low-hanging fruits so that if expected outcomes are not being achieved, there is room to stop and examine in real-time, how to make the efforts more powerful or re-calibrate to stay on the right trajectory. Furthermore, the variety of outcomes allows us to compare successful science efforts side by side, to determine if there are any patterns in influences through the SIF that most quickly or powerfully affect health outcomes. The SIF provides for an iterative process that continues to give; the assessment of the work can be on a continuum into the future. Thus, a retrospective assessment can be continued as a prospective monitoring for the foreseeable future. Perhaps some of our cases by now would have registered further impacts beyond what we found at the time we conducted these case studies and the respective programs can build on our findings to continue to monitor progress. New technologies such as machine learning and artificial intelligence can make the SIF assessments faster and easier, but human input will still be required to understand what the output means [75] and as other technologies become available, they can be leveraged as well. Several programs within CDC are beginning to use the SIF to measure program and public health impact. SIF was used by a CDC cooperative agreement recipient in assessing the uptake of CDC good

laboratory practice recommendations in biochemical genetic testing and newborn screening communities and developing plans to advance the impact [76]. In addition, a few findings using SIF have been published in peer-reviewed journals [77–79].

## Conclusion

In this paper, we present an approach to measuring science impact that goes beyond journal metrics. The initial development of the SIF and the case studies were based on CDC science, but has application beyond CDC. The SIF has flexibility that makes it feasible to assess retrospectively or monitor prospectively different efforts, those of established scientific programs, projects and research, specific scientific documents, such as publications and guidelines, and even individual scientist's body of work. The focus of public health is reducing morbidity and mortality with the goal of improving quality of life and wellbeing. The whole essence of the CDC science is to create what is actionable that would produce positive impact to keep people safe and healthy. It is important to know if what we think has potential of making an impact produces the anticipated impact. The SIF can serve as a framework for focusing and monitoring broader impact of science, beyond the impact of individual publications and products. With the SIF, a choice can be made as to what to monitor to show the broader impact of the science. What is unique about this is that the focus is not just on the projected impact or outcome rather on the effects that are occurring in real time with the recognition that the measurement field is complex. It can promote a culture change from assessing the impact of science primarily through journal metrics, to a more robust approach that captures qualitative data that measure the changes occurring because of science. Currently, it is rare to find a single source for evidence data. That may be more feasible for prospective monitoring as data sources can be determined in the planning stages of work, such as what systems to leverage to obtain data that substantiate impact. Moreover, it is for this reason that prospective monitoring using the SIF is considered easier than retrospective. However, once a retrospective assessment is done, future impacts can be tracked prospectively for that work, essentially SIF is evergreen in nature. Information generated from these assessments can be used to produce annual reports or other communication products to relay value added by science to the scientific community, policy makers and the public. The framework is broad enough and adaptable to address many areas of science. Almost anyone can tailor it to the work they do all that is needed is to define relevant key indicators for each of the domains of influence. Using the SIF will allow the translation of the value of science/research to the public in a simplified manner that is more likely to be of interest to them than peer-reviewed publication.

We are interested in the further dissemination and use of the SIF within the public health community or other venues.

## Acknowledgments

The authors would like to thank Tom Chapel for his perspective on how this framework complements the traditional CDC evaluation framework, Dr. Gerardo Garcia-Lerma for input in creating the Science Impact Framework figure and Dr. Betty Wong for review and comments on the case studies.

## Author Contributions

**Conceptualization:** Mary D. Ari, John Iskander, John Araujo, Christine Casey, Tanja Popovic.

**Data curation:** Mary D. Ari, John Araujo, Robert Swain.

**Investigation:** Mary D. Ari.

**Methodology:** Mary D. Ari, John Iskander, John Araujo, Christine Casey, Tanja Popovic.

**Project administration:** Mary D. Ari.

**Supervision:** Tanja Popovic.

**Validation:** Mary D. Ari, John Iskander, Christine Casey, John Kools, Bin Chen, Miriam Kelly.

**Writing – original draft:** Mary D. Ari.

**Writing – review & editing:** Mary D. Ari, John Iskander, John Araujo, Christine Casey, John Kools, Bin Chen, Robert Swain, Miriam Kelly, Tanja Popovic.

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
