## [Decision Letter · Decision Letter 0]

8 Oct 2020

PONE-D-20-25382

A Science Impact Framework to Measure Impact Beyond Journal Metrics

PLOS ONE

Dear Dr. Ari,

Thank you for submitting your manuscript to PLOS ONE. After careful consideration, we feel that it has merit but does not fully meet PLOS ONE’s publication criteria as it currently stands. Therefore, we invite you to submit a revised version of the manuscript that addresses the points raised during the review process.

We look forward to receiving your revised manuscript.

Kind regards,

Jim P Stimpson, PhD

Academic Editor

PLOS ONE

Journal Requirements:

Reviewers' comments:

Reviewer's Responses to Questions

**Comments to the Author**

1. Is the manuscript technically sound, and do the data support the conclusions?

Reviewer #1: Yes

2. Has the statistical analysis been performed appropriately and rigorously? 

Reviewer #1: N/A

3. Have the authors made all data underlying the findings in their manuscript fully available?

Reviewer #1: Yes

4. Is the manuscript presented in an intelligible fashion and written in standard English?

Reviewer #1: Yes

5. Review Comments to the Author

Reviewer #1: Is the manuscript technically sound, and do the data support the conclusions?

The authors' work is technically sound and literature review relatively thorough. The authors do not provide any quantitative study which is well justified in the paper. The qualitatitave study provided is clear and well presented.

Has the statistical analysis been performed appropriately and rigorously?

As the authors do not provide any quantitative study, this is not applicable.

Have the authors made all data underlying the findings in their manuscript fully available?

The authors do not provide any numerical data but the data in the qualitative study is clearly labelled and cited as appropriate.

Is the manuscript presented in an intelligible fashion and written in standard English?

The manuscript is very clear and well written.

- Overall a well argued, clear paper and a novel contribution to scientometrics.

- The authors provide a clear motivation and introduction for SIF.

- Although the authors briefly mention altmetrics later on in their discussion (ln 215) but may wish to briefly comment on them and their compatibility or their incompatibility with the SIF) in the intro/motivation for clarity.

- link to citation 21 appears to be broken (404 error).

- The 5 components of SIF and how different types of impact fit within the framework are well explained and illustrated.

- Agree that there is significant complexity and influences outside SIF to factor. However, sentence "For example, there may be other influences that may or may not work synergistically with the desired influence for the work under consideration" (lines 113-114) particularly difficult to read and parse. Could the authors rework or provide a more concrete example?

- Presumably "events or activities" (lns 132-133) can imply any measurable indicator from the framework including citations of synergistic and foundational work? Although the authors clarify this later (lns 166-167) it may help to reiterate this point earlier.

- lns 226-227 - Do the authors have a view on other US Federal works that have previously tried to capture some or all aspects of this information e.g. STAR Metrics (now Federal Reporter)?

- The authors' conclusion that a move away from simplistic "assessment of impact primarily through journal metrics" is fair and well substantiated by illustration. The view is shared by cited literature.

- A criticism of Research Excellence Framework is that collection and analysis of case studies is a very manual process which is incompatible with the growing volumes of scientific output requiring assessment (see Ravenscroft et al 2017. https://dx.doi.org/10.1371/journal.pone.0173152). SIF also heavily relies upon case studies and therefore may suffer similar criticism. With the concession that measurement of scientific impact is a complex and multi-faceted issue and that quantification of impact remains problematic, how do the authors respond to this criticism?

- The authors' concluding paragraphs and discussion of practical use of SIF are interesting and inspiring. However, it is clear that a lot of manual effort is likely to be involved in the creation, collection and evaluation of SIF case studies. Do the authors have a view on how modern Artificial Intelligence and Natural Language Processing approaches could help with these manual processes? A good relevant example could be McKeown et al. (2016) - https://doi.org/10.1002/asi.23612 who propose an automated approach for assignment of credit for development of scientific terms and tools over time by assessing appearance of technical terms in citation networks.

6. PLOS authors have the option to publish the peer review history of their article (what does this mean?). If published, this will include your full peer review and any attached files.

Reviewer #1: **Yes: **James Ravenscroft

---

## [Author Response · Author response to Decision Letter 0]

20 Nov 2020

- Although the authors briefly mention altmetrics later on in their discussion (ln 215) but may wish to briefly comment on them and their compatibility or their incompatibility with the SIF) in the intro/motivation for clarity.

The reason altimetric was mentioned in the discussion rather than the introduction is because we became aware of it after the development of the framework. The introduction discussed in details the frameworks we have reviewed during the development of our framework. MMWR and the CDC Library started collecting Altmetric data in 2014, that is when altimetric came to our attention. Altmetrics measures the volume of attention particular research has generated ONLINE. This data is mainly useful for identifying short term “reach” of specific publications. We see altimetric as serving as further resource that unearths attention to the publication being assessed. In a way we are already using altmetrics (information from media, policies, etc) but not in a formal way buy doing the professional altmetrics search way via Altmetric explorer or PlumX. It can be used for gathering data on the reach of a publication and allows a view of different venues where mention of the work appear and those can be analyzed for any evidence of impact. New tools like Altmetrics will continue to be discovered and used and they can be incorporated into the SIF as they mature. SIF is not a stagnant way of measuring science impact, but an umbrella that is broad and evolving

We have added statement in the paper indicating altimetric can serve as resource pointing to uptake of a publication. Lines 216 -218 - “but can provide data on the reach of publications and be a good resource in using the SIF.”

- link to citation 21 appears to be broken (404 error). 

We have obtained a new link. 

https://vdocuments.mx/reader/full/the-institute-of-medicine-what-makes-it-great-mediafilesabout-the-iom

- The 5 components of SIF and how different types of impact fit within the framework are well explained and illustrated.

- Agree that there is significant complexity and influences outside SIF to factor. However, sentence "For example, there may be other influences that may or may not work synergistically with the desired influence for the work under consideration" (lines 113-114) particularly difficult to read and parse. Could the authors rework or provide a more concrete example?

We added this text “Thus impacting the ability to achieve desired outcome positively or negatively.” At the heart of the sentence in lines 113-114 is the fact that we may not always get to the desired outcome, thus the need to track. This paper https://journals.plos.org/plosbiology/article?id=10.1371/journal.pbio.3000860 shows By paying attention to public perceptions of their publications, scientists can learn whether their research is stimulating positive scholarly and public thought. They can also become aware of potentially negative patterns of interest from groups that misinterpret their work in harmful ways, either willfully or unintentionally, and devise strategies for altering their messaging to mitigate these impacts. These impacts can go beyond messaging to actions.

- Presumably "events or activities" (lns 132-133) can imply any measurable indicator from the framework including citations of synergistic and foundational work? Although the authors clarify this later (lns 166-167) it may help to reiterate this point earlier.

Lines 129-130 indicated the significant events are based on the key indicators. The subsequent bullets describe how to make the linkages. We have added text that indicate synergistic work may be included. We are careful in highlighting this where we are describing identification of events and linkage as it may create the impression that one must always find synergy.

- lns 226-227 - Do the authors have a view on other US Federal works that have previously tried to capture some or all aspects of this information e.g. STAR Metrics (now Federal Reporter)?

Our understanding of Federal reporter is that it is a searchable database that captures research awards by federal agencies. It keeps counts by location as well as funding. It does not show the impact of the funded work, something we do using SIF.

- The authors' conclusion that a move away from simplistic "assessment of impact primarily through journal metrics" is fair and well substantiated by illustration. The view is shared by cited literature.

- A criticism of Research Excellence Framework is that collection and analysis of case studies is a very manual process which is incompatible with the growing volumes of scientific output requiring assessment (see Ravenscroft et al 2017. https://dx.doi.org/10.1371/journal.pone.0173152). SIF also heavily relies upon case studies and therefore may suffer similar criticism. With the concession that measurement of scientific impact is a complex and multi-faceted issue and that quantification of impact remains problematic, how do the authors respond to this criticism?

SIF, because it is a combination of quantitative and qualitative measures, is more labor intensive, but I think this is a reasonable tradeoff as we are seeking more than just numbers. It is fundamentally different and provides a richer basis for evaluation. SIF is not dependent on case studies as indicated by the reviewer, we used case studies for proof of concept. If you look at the published papers based on use of the SIF that we referenced lines 302-303, those are specific assessments not case studies. Case studies will be helpful for retrospective assessment of a large body of work covering several years, but as indicated in the paper future tracking is feasible precluding the need for case studies. Even though it is labor intensive now, as described in the paper, the review using SIF is akin to systematic review and if that has stood the test of time, we believe SIF will. Moreover, as indicated in our revision, as new ideas and technology become available, that will help make it more automated. See also Lines 84 to 86 “This is an abbreviated list of data sources; other resources as they become available may be used as needed.” We will continue to pay attention to evolving technologies, tools and resources that will minimize burden

- The authors' concluding paragraphs and discussion of practical use of SIF are interesting and inspiring. However, it is clear that a lot of manual effort is likely to be involved in the creation, collection and evaluation of SIF case studies. Do the authors have a view on how modern Artificial Intelligence and Natural Language Processing approaches could help with these manual processes? A good relevant example could be McKeown et al. (2016) - https://doi.org/10.1002/asi.23612 who propose an automated approach for assignment of credit for development of scientific terms and tools over time by assessing appearance of technical terms in citation networks.

Yes, it is labor intensive now, but as described in the paper, the review using SIF is akin to systematic review and if that has stood the test of time, we believe SIF will. Moreover, as indicated in our revision, as new ideas and technology become available, that will help make it more automated. See also Lines 84 to 86 “This is an abbreviated list of data sources; other resources as they become available may be used as needed.” We also added text to lines 296-298.

There is a concept called “human in the loop” AI systems, which basically means that no output from an AI or machine learning can stand as final without review by one or several subject matter experts. We have added text and this reference https://mitpress.mit.edu/books/artificial-unintelligence .So we think that AI could be part of the SIF “frontend”, but at the backend you need inputs from the program to understand what the output means.

---

## [Editor Report · Decision Letter 1]

9 Dec 2020

A Science Impact Framework to Measure Impact Beyond Journal Metrics

PONE-D-20-25382R1

Dear Dr. Ari,

We’re pleased to inform you that your manuscript has been judged scientifically suitable for publication and will be formally accepted for publication once it meets all outstanding technical requirements.

Kind regards,

Jim P Stimpson, PhD

Academic Editor

PLOS ONE
---

## [Editor Report · Acceptance letter]

11 Dec 2020

PONE-D-20-25382R1 

A Science Impact Framework to Measure Impact Beyond Journal Metrics 

Dear Dr. Ari:

I'm pleased to inform you that your manuscript has been deemed suitable for publication in PLOS ONE. Congratulations! Your manuscript is now with our production department. 

Kind regards, 

on behalf of

Dr. Jim P Stimpson 

Academic Editor

PLOS ONE